# The Effect of Torsional Vibration in Longitudinal–Torsional Coupled Ultrasonic Vibration-Assisted Grinding of Silicon Carbide Ceramics

**DOI:** 10.3390/ma14030688

**Published:** 2021-02-02

**Authors:** Yurong Chen, Honghua Su, Jingyuan He, Ning Qian, Jiaqing Gu, Jiuhua Xu, Kai Ding

**Affiliations:** 1National Key Laboratory of Science and Technology on Helicopter Transmission, College of Mechanical and Electrical Engineering, Nanjing University of Aeronautics and Astronautics, Nanjing 210016, China; chenyr@amt-nuaa.com (Y.C.); hjy8817@126.com (J.H.); n.qian@nuaa.edu.cn (N.Q.); 18252583752@163.com (J.G.); jhxu@nuaa.edu.cn (J.X.); 2School of Mechanical Engineering, Jiangsu University of Technology, Changzhou 213001, China; dingkai@jsut.edu.cn

**Keywords:** ultrasonic, longitudinal–torsional coupled, actual amplitude, grinding force, surface roughness

## Abstract

Rotary longitudinal–torsional coupled ultrasonic vibration-assisted grinding (LTUAG) is a new manufacturing method that can improve the grinding ability of silicon carbide ceramics. However, compared with longitudinal ultrasonic vibration-assisted grinding (LUAG), the role of torsional vibration in the grinding process is unclear. In this study, an effective method for measuring longitudinal–torsional coupled ultrasonic vibration amplitude and an experimental setup for measuring actual amplitude during grinding are proposed. The trajectory of the abrasive grains under the same grinding parameters and the same longitudinal amplitude during LTUAG and LUAG are analysed. Ultrasonic amplitude curves under the condition of tool rotation are then measured and analysed. Finally, the effect of torsional vibration on grinding force and surface roughness under the same grinding conditions is explained. Experimental analysis shows that the introduction of torsional vibration has little effect on the trajectory length and does not change the number of interference overlaps between abrasive grain tracks. Torsional vibration will only increase the cutting speed during grinding and reduce the undeformed chip thickness, which will reduce the grinding force and improve the surface roughness of LTUAG.

## 1. Introduction

As a typical hard and brittle material, silicon carbide (SiC) ceramics are widely used in optical reflectors, bearings, molds, and some parts in aeronautics and astronautics fields. However, SiC ceramics suffer from easy crack generation and fast tool wear during conventional grinding (CG) due to the material properties of high hardness and high brittleness, which severely impact both the process efficiency and the process costs of grinding SiC.

In recent years, rotary longitudinal ultrasonic vibration-assisted grinding (LUAG) has been widely used in the machining of hard and brittle materials. To date, many studies have proved LUAG is an effective method for grinding SiC [1], ceramic matrix composites [2,3], and optical glass [4,5]. The vibration has different effects in grinding under different grinding modes with ultrasonic vibration. Compared with conventional end-face grinding, the vibrating abrasive grains in ultrasonic vibration-assisted grinding continuously impact the workpiece’s surface; the vibrations generate a micro-broken surface, which improves the material removal rate [6]. Correspondingly, contact and separation between the workpiece and the tool reduce the average grinding force [6,7], reducing the edge chipping of workpieces [8]. However, the surface roughness of LUAG is worse than that of CG because of the impaction [1]. For ultrasonic vibration-assisted side-face grinding, the trajectories of abrasive grains intersect and overlap each other, reducing the surface roughness of the workpiece’s surface [9]. At the same time, the length of the abrasive grain track with vibration is also larger than that of CG, which can reduce the maximum undeformed chip thickness of a single abrasive grain and the grinding force [10], extending the life of the tool [11]. To date, considerable progress has been made in the investigation of LUAG in the material removal mechanism and process optimisation, among other aspects. To further improve the process effect of ultrasonic vibration-assisted grinding on hard and brittle materials, some studies have proposed the use of longitudinal–torsional coupled vibration instead of longitudinal vibration for grinding.

Generally, there are two main investigations to be made into rotary longitudinal–torsional coupled ultrasonic vibration-assisted grinding (LTUAG), namely the design of the LTUAG device and the optimisation of the LTUAG process. In terms of the design of the LTUAG device, the amplitude ratio of torsional-to-longitudinal vibration is a critical parameter for LTUAG and is mainly integrated through the structure to achieve the vibration modal conversion from longitudinal vibration to torsional vibration [12,13,14]. In terms of optimising the LTUAG process parameters, compared to longitudinal ultrasonic vibration, longitudinal–torsional coupled ultrasonic vibration can not only reduce more than 50% of the axial force when drilling hard and brittle materials [15], but can also improve the material removal rate [16]. In addition to the machining of hard and brittle materials, longitudinal–torsional coupled ultrasonic vibration also has certain advantages in reducing the cutting force, improving the surface quality, and extending tool life in the drilling [17] and milling [18] of metal materials.

In summary, rotary longitudinal–torsional coupled ultrasonic vibration-assisted machining shows significantly superior performance than longitudinal ultrasonic vibration-assisted machining. However, the torsional vibration vibrates along with the circular direction of the tool, and the torsional amplitude of the tool is difficult to measure. Therefore, the method of finite element simulation is usually used to obtain the value of the amplitude ratio. An undeterminable error exists between the amplitude ratio obtained by finite element simulation and the real amplitude ratio of the LTUAG device, and this will affect the analysis of the machining results.

In terms of process, the axial amplitudes of LTUAG and LUAG in these studies are static amplitudes (SAs) during the comparative experiment, which are measured before machining and assumed to remain unchanged during the process. However, the actual amplitudes (AAs) during grinding are affected by the grinding force, which causes the difference in the longitudinal amplitude of the two grinding methods and affects the reliability of the experimental analysis. At the same time, compared with the SA, the AA in grinding is difficult to measure in real time, owing to the influence of tool rotation and cutting fluid. Therefore, solving the problem of the real-time measurement of the AA in grinding is the key to ensuring the consistency of the longitudinal amplitude of the two grinding methods. Some studies have also analysed the difference in grinding force and surface quality between LTUAG and LUAG but have not clearly explained the main reason for the change in grinding force and surface roughness due to the addition of torsional vibration.

Given all the aforementioned issues, a method of measuring the amplitude of longitudinal–torsional coupled ultrasonic vibration is proposed in this study, and the abrasive trajectories of LUAG and LTUAG are analysed theoretically. Then, an apparatus is built for measuring the AA during LTUAG and LUAG in real time. Thereafter, the influences of grinding parameters on side grinding forces and surface roughness are discussed by comparing the LTUAG and LUAG of SiC ceramics, thus clarifying the effect of torsional vibration on the grinding force and roughness by analysing factors such as the maximum undeformed chip thickness of a single abrasive grain, the trajectory length, and overlapping trajectories of adjacent abrasive grains.

## 2. Kinematic Analysis of LUAG and LTUAG

The trajectories of a single abrasive under LUAG and LTUAG conditions are described by Equations (1) and (2), respectively.
(1)$$\{\begin{array}{c} X_{L}=R\cos(\omega t)+v_{w}t\\ Y_{L}=R\sin(\omega t)\\ Z_{L}=A_{L}\sin(2\pi ft) \end{array}$$
(2)$$\{\begin{array}{c} X_{LT}=R\cos(\omega t+\theta \sin(2\pi ft))+v_{w}t\\ Y_{LT}=R\sin(\omega t+\theta \sin(2\pi ft))\\ Z_{LT}=A_{L}\sin(2\pi ft+\phi ) \end{array}$$
where *R* is the radius of the tool (*ω* = 2π*n*/60), *n* is the rotating speed of the tool, *v_f_* is the feed speed of the tool, *f* is the frequency of the ultrasonic tool holder, *A_L_* is the amplitude of the longitudinal vibration, *θ* is the torsional angle, and *φ* is the phase difference between the longitudinal vibration and torsional vibration.

Figure 1a shows the transducer of the longitudinal–torsional coupled ultrasonic tool holder, which has eight spiral grooves. Each spiral groove is 10 mm long by 1 mm wide and 2 mm deep and has a helix angle of 45°. The amplitude ratio of the torsional-to-longitudinal vibration is measured by a PSV laser doppler sensor. The method of the measurement is shown in Figure 1b. The square tool is clamped on the front of the transducer; the laser sensor is parallel to the plane of the square tool. The PSV laser doppler sensor can measure the vibration modal and amplitude of the tool plane, and the amplitude ratio can be calculated by the amplitude in the X, Y, and Z directions. When the resonant frequency of the longitudinal–torsional ultrasonic oscillator is 23.73 kHz, the end-face vibration mode is longitudinal, which is vertical to the end face within the vibration period. When the end face of the square tool vibrates to the highest position, the measured vibration-modal cloud image is red, while the vibration-modal cloud image at the lowest position is green (Figure 1c). The side face vibration modal is a torsional vibration with the side centre line as the axis of symmetry (Figure 1d). When the output power of the ultrasonic power supply is 61%, the torsional amplitude is 2.73 μm and the longitudinal amplitude is 5.79 μm (Figure 1e). The amplitude ratio of torsional-to-longitudinal vibration is 0.472.

Figure 2 illustrates the trajectory of a single abrasive under LUAG and LTUAG, when the radius of the tool is 4 mm, grinding speed is 1.25 m/s, feed speed is 4000 mm/min, frequency is 23.73 kHz and longitudinal amplitude is 10 μm by combining Equations (1) and (2) and the value of amplitude ratio. The trajectory of LUAG is a periodic sine curve, whereas the trajectory of LTUAG is a periodic distortion curve.

## 3. Experimental Details

The comparative experiments of LUAG and LTUAG were conducted on a DMG Ultrasonic 20 linear machining centre (DMG, Bavaria, Germany), and the grinding method was side-grinding (Figure 3). The experimental setup was composed of an ultrasonic tool holder, primary coil device, ultrasonic power supply, oscilloscope (R&S, Munich, Germany), eddy current sensor (Micro-Epsilon, Bavaria, Germany), sensor controller, Kistler 9272 dynamometer (Kistler, Winterthur, Switzerland), charge amplifier, and computer (Figure 4). The workpiece was made of SiC ceramic and bonded to the adapter plate by paraffin wax. The adapter plate was fixed on the Kistler 9272 dynamometer by screws. The charge amplifier of the dynamometer was connected to a computer to collect the grinding forces during the grinding process. The eddy current sensor was connected to the sensor controller, and it moved with the spindle and tool to measure the longitudinal amplitude of the tool end face during LTUAG and LUAG. The ultrasonic tool holders of LUAG and LTUAG are shown in Figure 4a,b, respectively.

The tool is an electroplated diamond tool with a diameter of 8 mm. The diameter of each diamond abrasive is 151 μm. The size of the SiC ceramic workpiece is 50 mm × 50 mm × 10 mm. The mechanical properties of the SiC ceramic are listed in Table 1.

To analyse the influences of grinding parameters on the grinding forces and surface roughness, single-factor experiments were conducted during LTUAG and LUAG. The specific process parameters are listed in Table 2. In addition, the average grinding forces and roughness value were used to analyse the results.

## 4. Measurement Method of Actual Amplitude

During the grinding, the eddy current sensor moved with the spindle and was always parallel to the end face of the electroplated diamond tool. The measurement range of the eddy current sensor was 0–0.5 mm, the sampling frequency was 100 kHz, and the measurement accuracy was 0.5 μm. In this process, the eddy current sensor could only measure the distance on metal; thus, the magnetic field of the eddy current sensor passed through the diamond abrasive grains and measured the distance of the tool’s nickel-plated surface. The oscilloscope collected the voltage originating from the eddy current sensor controller, which represents the distance between the sensor and the tool end face. The amplitude of the rotating tool can be express by
(3)A=U*η
where *A* is the axial amplitude of the rotating tool, *U* is the output voltage of the eddy current sensor, and *η* is the amplitude-to-voltage ratio (i.e., 50 μm/V).

In the grinding process, the grinding force deflects the tool. In order to study the influence of the grinding force on the tool deflection, the static analysis of the ultrasonic vibrator is analysed by the finite element analysis method. Generally, when there was no obvious defect to the workpiece, the grinding force of the grinding modes described in this research was less than 20 N. As shown in Figure 5, with the increase in the grinding force from 4 N to 20 N, the deformation increases with the increase in the grinding force. The maximum deformation of the tool is 11.6 μm under a grinding force of 20 N (Figure 6).

The tool deflection changes the distance (*d*) and angle (*α*) between the eddy current sensor and the end face of the tool (Figure 7) and is much smaller than the measurement range of the eddy current sensor, which does not affect the value and accuracy of the measurement. Meanwhile, tool deflection is similar to axial run-out of the tool (Figure 8); both of them are low-frequency signals, which can be filtered when analysing actual amplitude. Therefore, the tool deflection does not affect the actual amplitude measurement and analysis.

To study the effects of the relative position of the eddy current sensor and the end face of the tool on the AA, the distance between the centre of the eddy current sensor and the tool centre was set at 1.5, 2.25, and 3 mm, as shown in Figure 9. When the rotational speed was 12,000 rpm, the amplitude of the distance at 1.5, 2.25, and 3 mm was 5.84, 6.09, and 5.74 μm, respectively. The maximum error for all three of the amplitudes was 0.35 μm, which was smaller than the measurement error of the eddy current sensor. The preceding research shows that, regardless of the relative position of the eddy current sensor and the tool, the eddy current sensor can effectively measure the amplitude of the end face of the tool. It also indicates that this method for measuring the AA exhibits universal applicability.

Figure 8a presents the vibration curve of the tool with rotation and without load when the tool grinding speed was 5.03 m/s (12,000 r/min), the output power of the ultrasonic power supply was 45% and the maximum amplitude of the electroplated grinding tool was 27.2 μm. By filtering the vibration curve in Figure 8a, the curve can be decomposed into two curves with different frequencies, as shown in Figure 8b,c. In Figure 8b, the amplitude of axial run-out is 15.62 μm and the frequency is 200 Hz. In Figure 8c, the amplitude of ultrasonic vibration is 5.8 μm and the frequency is 25.25 kHz, which is similar to the SA of 5.9 μm (Figure 10). According to the above analysis, the amplitude of the tool with ultrasonic vibration is the sum of the axial run-out of the tool and the ultrasonic vibration SA.

Figure 11 shows the actual amplitude curve when the grinding speed was 5.03 m/s, the feed speed was 1100 mm/min, and the depth of cut was 29 μm. The amplitude after filtering was 4.6 μm, whereas the amplitude without load was reduced by 1.2 μm. The preceding research shows that this method of measuring amplitude can effectively measure the AA of the grinding process in real time.

To study the effect of torsional vibration in LTUAG, it was ensured that the resonant frequency and the longitudinal amplitude were the same by changing the overhang length of the tool and the output power of the ultrasonic power supply, respectively, in the comparative experiment of LUAG and LTUAG. In the LUAG experiment, when the overhang length of the tool was 26 mm, the resonance frequency of the longitudinal ultrasonic tool holder was 24.4 kHz. In the experiment of LTUAG, when the overhang length of the tool was 22 mm, the resonance frequency of the longitudinal–torsional coupled ultrasonic tool holder was 24.4 kHz. The ultrasonic vibration parameters of the LUAG and LTUAG experiments are listed in Table 3.

## 5. Results and Discussion

### 5.1. Grinding Force of LTUAG and LUAG

A comparison of the cutting force between LUAG and LTUAG with the grinding parameters changed is shown in Figure 12. In the experiments, *F_t_* and *F_n_* refer to the tangential and normal grinding forces in LUAG and LTUAG, respectively. As shown in Figure 12, no matter how the grinding parameters change, the grinding force produced by LTUAG is always smaller than that produced by LUAG. When the depth of cut was 10 μm and the feed speed was 400 mm/min, with an increase in the grinding speed from 1.25 to 8.35 m/s, the grinding force decreased in both LUAG and LTUAG. The tangential and normal grinding forces of LTUAG decreased by 1.3 to 38.5% and 14.8 to 37.8% compared to those of LUAG, and the reduction rate of the grinding force on LTUAG to LUAG decreased with the increase in grinding speed (Figure 12a). Figure 12b shows the effect of the feed speed on the grinding forces in LUAG and LTUAG. When the grinding speed was 1.25 m/s and the depth of cut was 10 μm, the feed speed changed from 100 mm/min to 1000 mm/min, and the grinding forces of LUAG and LTUAG increased with the increase in feed speed. The tangential and normal grinding forces of LTUAG decreased by 7.2 to 45.3% and 17.5 to 58.9% compared to the grinding force of LUAG, and the reduction rate of the grinding force on LTUAG to LUAG increased with the increase in grinding speed. When the grinding speed was 1.25 m/s and the feed speed was 400 mm/min, the grinding depth changed from 5 to 20 μm; as the grinding depth increased, the grinding forces of LUAG and LTUAG both increased, and compared to LUAG, the tangential and normal grinding force for LTUAG decreased by 20.4 to 38.5% and 21.7 to 33.1%. However, the change of grinding depth has little effect on the reduction rate of the LTUAG grinding force (Figure 12c).

The ultrasonic grinding force is related to the undeformed chip thickness [19,20,21], and the maximum undeformed chip thickness *a_Ugmax_* can be expressed as Equation (4):(4)$$a_{Ug\max}=\left[\frac{v_{w}}{v_{s}}a_{p}\right]{\left[\frac{N_{d}bl_{Us}}{2}\right]}^{-1}$$
where *N_d_* is the number of dynamic effective abrasive grains in the grinding arc zone, *b* is the width of the grinding arc zone and *l_Us_* is the abrasive trajectory length.

In comparison to LUAG, due to the vibration of the tool in the circumferential direction during the LTUAG, the two kinds of process methods have different abrasive trajectory lengths within the grinding arc zone. The trajectory lengths of LUAG and LTUAG are shown in Equations (5) and (6), respectively:(5)$$l_{LUAG}=Rf/v_{s}\arccos\frac{R-a_{p}}{R}\int_{0}^{1/f}\sqrt{dX_{L}^{2}+dY_{L}^{2}+dZ_{L}^{2}}dt$$
(6)$$l_{LTUAG}=Rf/v_{s}\arccos\frac{R-a_{p}}{R}\int_{0}^{1/f}\sqrt{dX_{LT}^{2}+dY_{LT}^{2}+dZ_{LT}^{2}}dt$$

By combining with Equations (1) and (2), we can get the following:(7)$$l_{LUAG}=Rf/v_{s}\arccos\frac{R-a_{p}}{R}\int_{0}^{1/f}\sqrt{4\pi^{2}f^{2}A_{L}^{2}\cos^{2}(2\pi ft)-2v_{s}v_{w}\sin\omega t+v_{s}^{2}+v_{w}^{2}}dt$$
(8)$$l_{LTUAG}=Rf/v_{s}\arccos\frac{R-a_{p}}{R}\int_{0}^{1/f}\sqrt{\begin{array}{l} 4\pi^{2}f^{2}A_{L}^{2}\cos^{2}(2\pi ft+\phi )-(2v_{s}v_{w}+4Rv_{w}\pi f\theta \cos(2\pi ft))\sin(\omega t+\theta \sin(2\pi ft))\\ +{\left(v_{s}+2R\pi f\theta \cos(2\pi ft)\right)}^{2}+v_{w}^{2} \end{array}}dt$$
where *θ**R =*
*A_T_*, *A_T_* is the amplitude of the torsional vibration. Through Equations (7) and (8), we know that the abrasive trajectory of ultrasonic vibration-assisted grinding is related to the grinding parameters of the grinding speed and feed speed and has nothing to do with the grinding depth. The change rules of *L_LUAG_* and *L_LTUAG_* with grinding speed and feed speed based on the grinding parameters of Table 1 are shown in Figure 13.

Figure 13 reveals that when the grinding speed was 1.25 m/s, the trajectory length of the longitudinal–torsional composite vibration abrasive grains in one vibration period is larger than the motion trajectory of the longitudinal vibration abrasive grains, and the difference in trajectory length ratio was 13.6%. When the grinding speed was larger than 3.35 m/s, the difference between the two vibration trajectory lengths was less than 0.3%. For the feed rate, the change of feed speed hardly affected the trajectory-length difference between longitudinal–torsional vibration and longitudinal vibration. Therefore, when the longitudinal amplitude was the same, the trajectory lengths of the abrasive grains during LTUAG were almost the same as the trajectory lengths of LUAG.

For the grinding speed, compared to LUAG, the grinding speed of LTUAG varied periodically (Figure 14). The periodically changing grinding speed of LTUAG changes the maximum undeformed chip thickness. The torsional vibration can also enlarge the actual grinding speed of the abrasive, thus facilitating material removal in brittle mode with a micro fracture chip [22]. However, when the grinding speed increases, the grinding speed different ratio decreases (Figure 15), and the grinding speed of LTUAG is close to that of LUAG. At this time, the role of torsional vibration in LTUAG decreases, and its grinding force approaches the grinding force of LUAG.

### 5.2. Surface Roughness of LTUAG and LUAG

The surface roughness of LUAG and LTUAG is measured with the MarSurf PS1 roughness tester. The measurement direction is perpendicular to the feed speed direction and the average value of the roughness curvature at three different positions of the grinding surface is measured. The effect of grinding parameters to surface roughness under the conditions of the same longitudinal amplitude of LUAG and LTUAG is shown in Figure 16. The surface roughness produced by LTUAG is shown to be less than that produced by LUAG. Furthermore, the results indicate that the surface roughness decreased with increased grinding speed; as the feed speed and grinding depth increased, the roughness became worse. Compared to LUAG, when the grinding speed changed from 1.25 to 8.35 m/s and the feed speed changed from 100 to 1000 mm/min, the roughness of LTUAG reduced by 1.36 to 8.3% and 6.75 to 9.97%, respectively, as shown in Figure 16a,b. When the grinding depth changed from 5 to 20 μm, the value of the reduction rate had a slight increase that was within 2% (Figure 16c).

Compared to conventional grinding, studies [23,24] have shown that the interference between the trajectories of different abrasives is the main reason for improving the roughness of ultrasonic vibration-assisted grinding. Paths 1, 2, 3, and 4 are the trajectories of four adjacent abrasive grains (Figure 17). Compared with the trajectory of LUAG, the trajectories of the adjacent abrasive grains overlapped 12 times in one vibration period; there was no difference in the number of overlaps between LTUAG and LUAG trajectories. Therefore, the decrease in surface roughness of LTUAG was not caused by a change in the number of interference abrasive grains. The analysis in the previous section indicates that the torsional vibration of LTUAG changed the grinding speed and the maximum undeformed chip thickness, which was also the main reason for reducing the surface roughness of LTUAG.

## 6. Conclusions

(1)A new method for measuring the amplitude ratio of torsional to longitudinal vibration by using a PSV laser doppler sensor and a square tool is proposed in this study. The longitudinal–torsional coupled vibration ultrasonic transducer has eight spiral grooves, with each spiral groove measuring 10 mm long by 1 mm wide and 2 mm deep. The spiral groove has a helix angle of 45 degrees, and the amplitude ratio is 0.472.(2)A method for the real-time measurement of the AA during grinding is proposed. This method can be free from the interferences and influences of external factors, such as tool rotation and cutting fluid. With the experimental setup, the experiments show that the proposed method is effective for AA measurement, and can ensure the consistency of the longitudinal amplitude of LTUAG and LUAG.(3)When the longitudinal amplitudes of LTUAG and LUAG are the same, the torsional vibration of LTUAG will not increase the trajectory length of the single abrasive. However, the torsional vibration can increase the grinding speed of LTUAG, which reduces the undeformed chip thickness of LTUAG, and the grinding force of LTUAG is smaller than that of LUAG.(4)Accompanied by a reduction in the undeformed chip thickness, the surface roughness of LTUAG has been effectively improved. Compared with LUAG, the roughness of LTUAG reduced by 1.36 to 9.97%. With the increase in grinding speed and feed speed, the roughness of LTUAG approached the roughness produced by LUAG because the grinding speed of LTUAG was close to that of LUAG.

## Figures and Tables

**Figure 1 materials-14-00688-f001:**
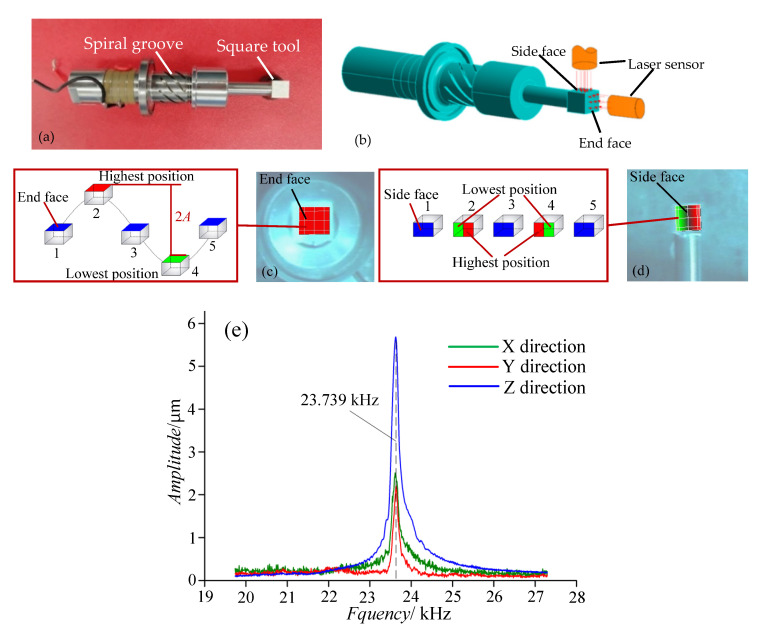
Longitudinal–torsional coupled ultrasonic transducer. (**a**) Transducer of the longitudinal–torsional coupled ultrasonic tool holder. (**b**) Method of the measurement. (**c**) Longitudinal vibration. (**d**) Torsional vibration. (**e**) Amplitude-frequency curve.

**Figure 2 materials-14-00688-f002:**
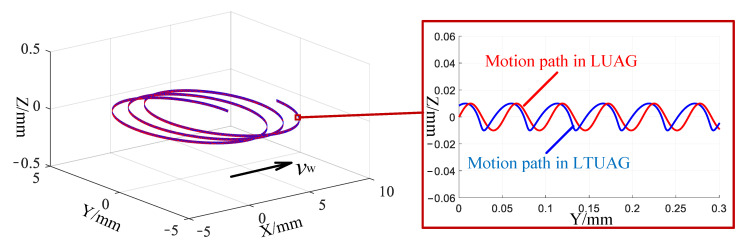
Trajectory of single abrasive in both LUAG and LTUAG.

**Figure 3 materials-14-00688-f003:**
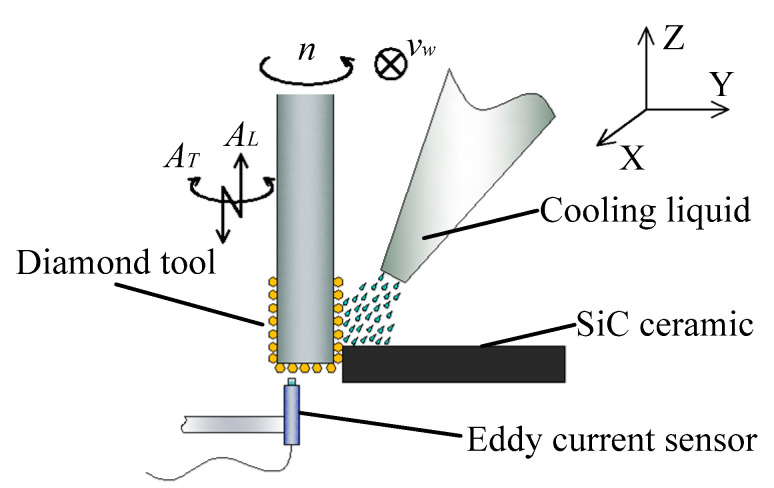
Schematic of the apparatus.

**Figure 4 materials-14-00688-f004:**
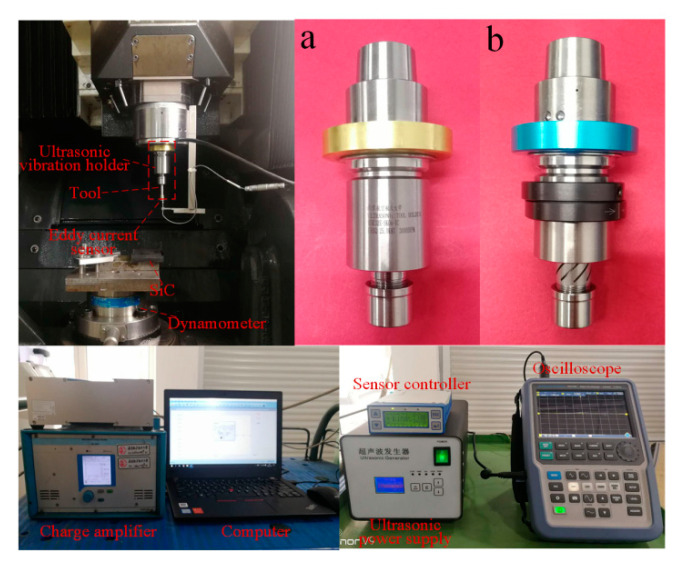
Photographs of the experimental setups. (**a**) Longitudinal ultrasonic tool holder. (**b**) Longitudinal–torsional coupled ultrasonic tool holder.

**Figure 5 materials-14-00688-f005:**
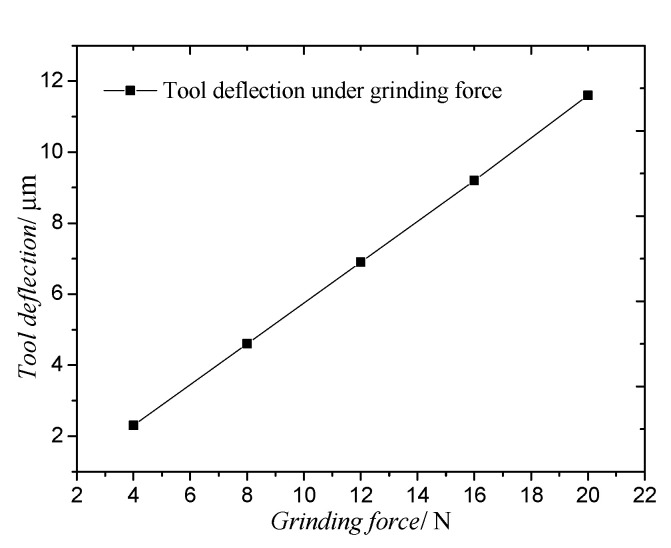
Tool deflection under grinding force.

**Figure 6 materials-14-00688-f006:**
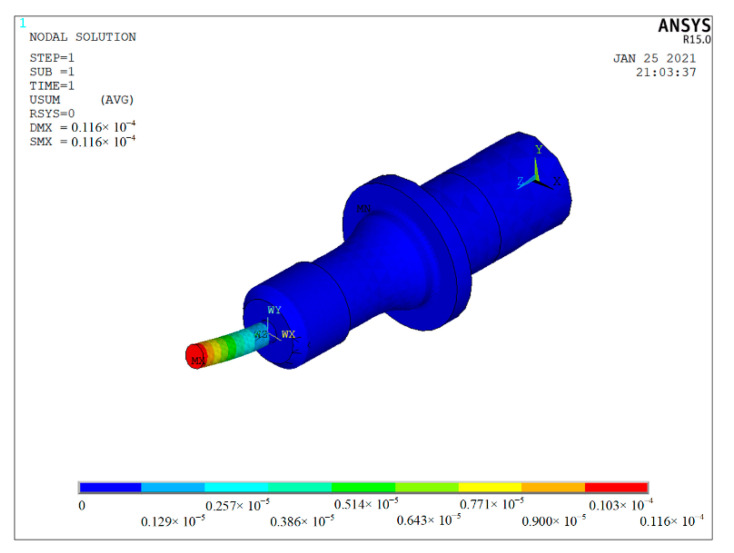
The cloud image of ultrasonic vibrator deflection.

**Figure 7 materials-14-00688-f007:**
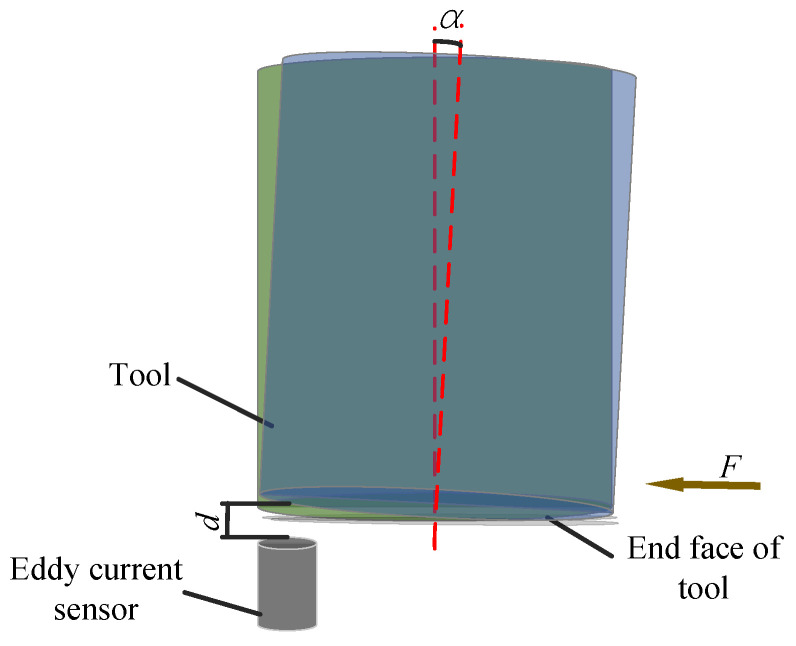
Schematic diagram of tool deflection.

**Figure 8 materials-14-00688-f008:**
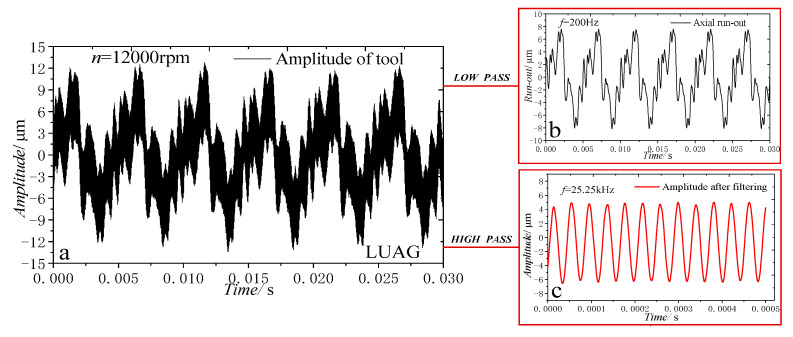
Ultrasonic amplitude of the tool with rotation: (**a**) overall curve, (**b**) axial run-out, (**c**) actual amplitude.

**Figure 9 materials-14-00688-f009:**
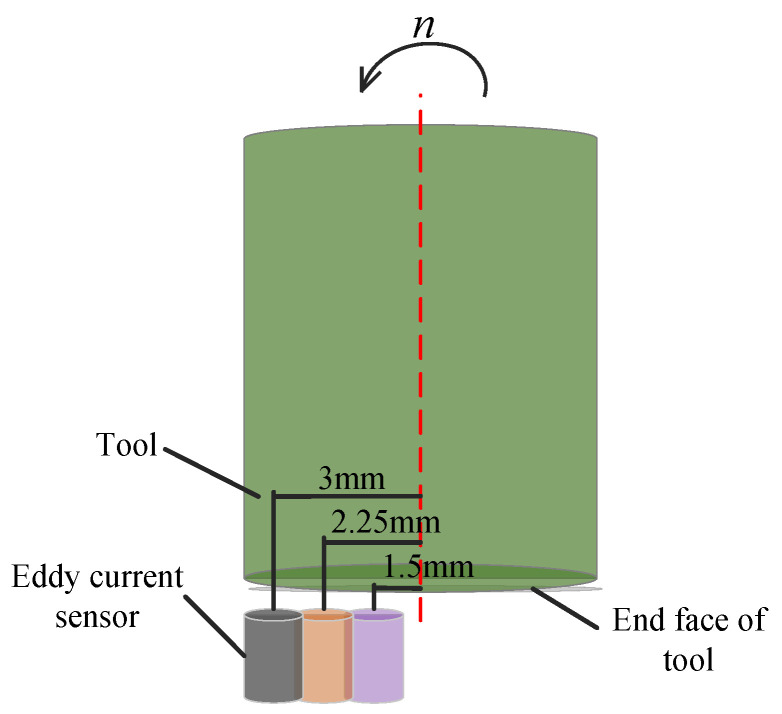
Different position of eddy current sensor.

**Figure 10 materials-14-00688-f010:**
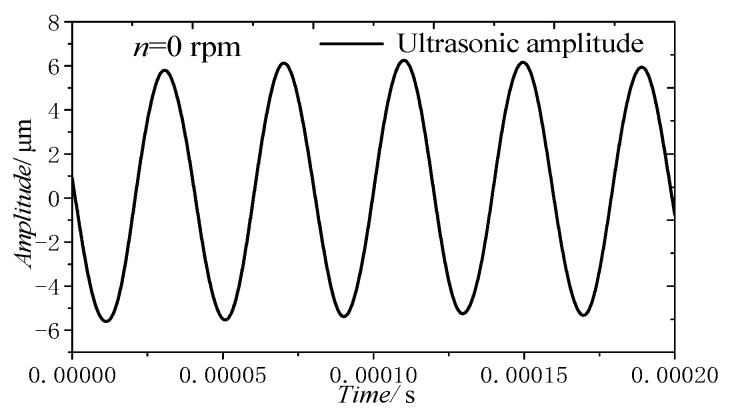
Ultrasonic vibration static amplitude.

**Figure 11 materials-14-00688-f011:**
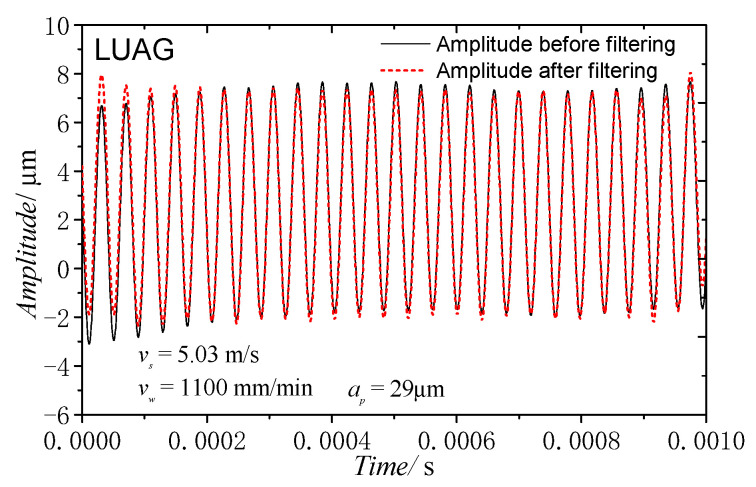
Amplitude curve under grinding force of ultrasonic vibration-assisted side-face grinding.

**Figure 12 materials-14-00688-f012:**
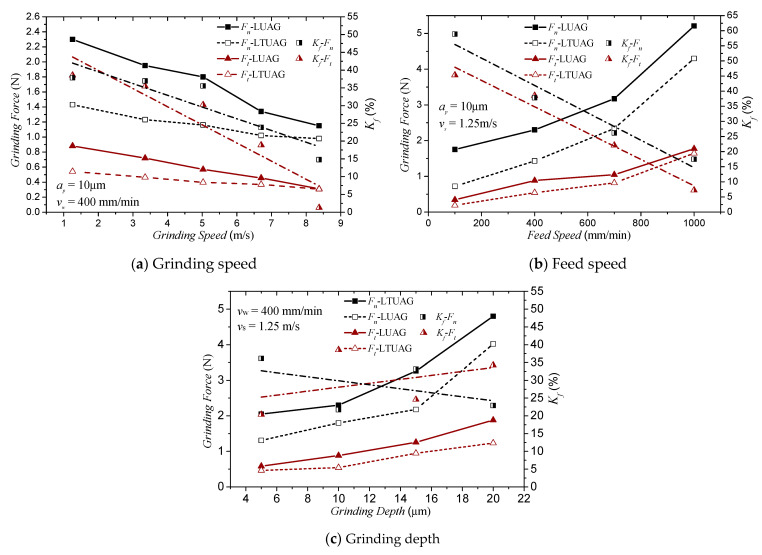
Effect of grinding parameters on grinding force (*K_f_* is the reduction rate of the grinding force produced by LTUAG compared to the grinding force produced by LUAG, *K_f_* = (*F_LUAG_* − *F_LTUAG_*) × 100%/*F_LUAG_*).

**Figure 13 materials-14-00688-f013:**
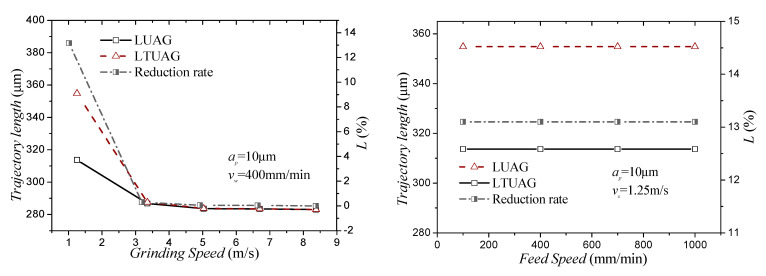
The trajectory length of LTUAG and LUAG.

**Figure 14 materials-14-00688-f014:**
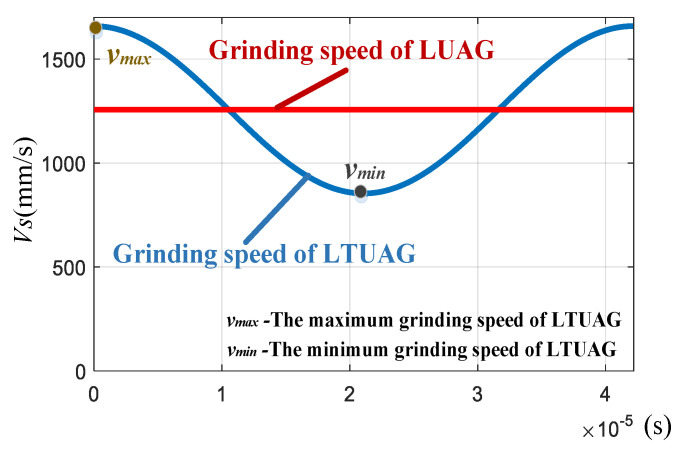
Grinding speed of LTUAG and LUAG.

**Figure 15 materials-14-00688-f015:**
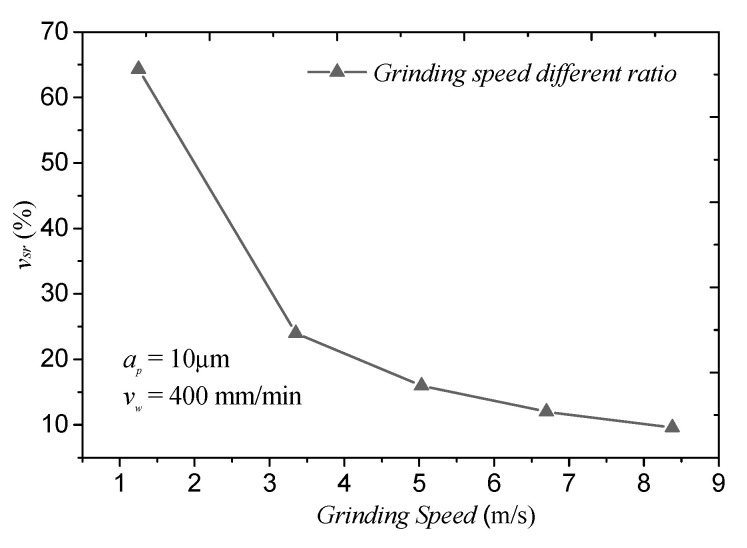
Grinding speed different ratio (*v_sr_* is the grinding speed different ratio, *v_sr_* = (*v_max_* − *v_min_*) × 100%/*v_s_*).

**Figure 16 materials-14-00688-f016:**
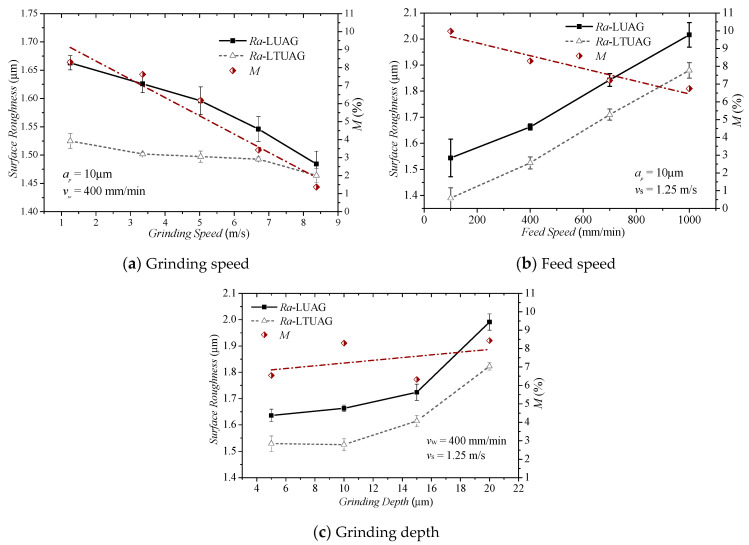
Effect of grinding parameters on surface roughness (*M* is the reduction rate of the surface roughness produced by LTUAG compared to the surface roughness produced by LUAG, *M* = (*Ra_LUAG_* − *Ra_LTUAG_*) × 100%/*Ra_LUAG_*).

**Figure 17 materials-14-00688-f017:**
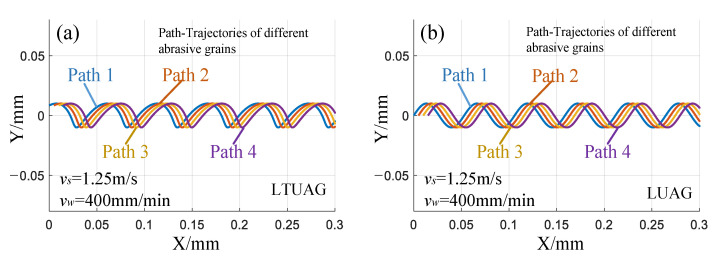
Overlapping trajectories of adjacent abrasive grains. (**a**) Trajectories of LTUAG, (**b**) Trajectories of LUAG.

**Table 1 materials-14-00688-t001:** Properties of SiC ceramic.

Properties	Value	Unit
SiC content	≥98	%
Density	3.15	g/cm^3^
Fracture toughness	3.2	MPa·m^1/2^
Vickers hardness	28	GPa
Young’s modulus	4.1 × 10^5^	MPa

**Table 2 materials-14-00688-t002:** Parameters of the grinding process.

Parameters	Value
Grinding speed *v_s_*/(m/s)	1.25, 3.35, 5.03, 6.7, 8.38
Feed speed *v_w_*/(mm/min)	100, 400, 700, 1000
Grinding depth *a_p_* (μm)	5, 10, 15, 20
Radial depth of cut *a_e_* (mm)	2

**Table 3 materials-14-00688-t003:** Ultrasonic vibration parameters of LUAG and LTUAG.

Parameters	Value
Overhang length of the tool (mm)	LUAG: 26 LTUAG: 22
Frequency (kHz)	24.4
Actual amplitude (μm)	LUAG: 6 (*A_L_*) LTUAG: 6 (*A_L_*), 2.8 (*A_T_*)

## Data Availability

Data sharing is not applicable to this article.

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
