# Peer review of "The Effect of Torsional Vibration in Longitudinal–Torsional Coupled Ultrasonic Vibration-Assisted Grinding of Silicon Carbide Ceramics"

_materials, 2021, doi:10.3390/ma14030688_

Round 1
Reviewer 1 Report
- The value sets corresponding to the ae and ap in Table 1 seem to be displaced and should be corrected.
- The introduction provides a good background of the ultrasonic assisted grinding, however, the more important fact than "the longitudinal and/or torsional vibrations" is being missed, which is "what improvements are introduced by applying ultrasonic vibrations in comparison to conventional grinding?". This question should be answered and if of great relevance for a academic of technical reader.
- Some more previous studies could be cited, which are not less relevant than the referenced literature:
- Ultrasonic assisted grinding of ceramics, Journal of Materials Processing Technology, Volume 62, Issue 4, December 1996, Pages 287-293
- Ultrasonic vibration assisted grinding of bio-ceramic materials: an experimental study on edge chippings with Hertzian indentation tests, The International Journal of Advanced Manufacturing Technology volume 86, pages3483–3494(2016)
- Energy aspects and workpiece surface characteristics in ultrasonic-assisted cylindrical grinding of alumina–zirconia ceramics, International Journal of Machine Tools and Manufacture, Volume 90, March 2015, Pages 16-28
- Advances in Ultrasonic Assisted Grinding of Ceramic Materials, Advances in Science and Technology (Volume 45), Pages: 1711-1716
- The effect of different grinding parameters are suitably covered and presented. The same quantitative approach should be also conducted for the case of conventional grinding (without ultrasonic vibrations). Maybe this even helps to demonstrate further improvements because of ultrasonic vibrations.
- Although the amplitude measurement is conducted in axial tool direction, owing to the rather small tool diameter, it could expected (especially for large depths of cut and feed velocities) that the tool deflection induced uncertainties in the amplitude measurement with the eddy-current sensor. Even a small shift in the measurement point on a grinding tool with a rather rough grain size (151µm) could cause measurement errors which might be much larger than the measured values. (How) Did the authors compensate for this effect?
Reviewer 2 Report
The reviewer comments of the paper «Effect of torsional vibration in longitudinal-torsional coupled ultrasonic vibration assisted grinding of silicon carbide ceramics»- Reviewer
The authors presented an article «Effect of torsional vibration in longitudinal-torsional coupled ultrasonic vibration assisted grinding of silicon carbide ceramics». In general, the article is interesting and written at a good scientific level. However, there are several points in the article that require further explanation.
Comment 1:
The introduction should be expanded. Try to reinforce the relevance of your research. It is important to more clearly define why they chose silicon carbide ceramics for research. What are the difficulties when grinding such materials? Why is the proposed method more relevant for this? Thus, add a paragraph with analysis. Show more articles especially from the last 5 years.
It is necessary to more clearly define the "white spots". That has not been previously investigated by other scientists. At the end of the introduction, state a clear research objective.
Comment 2:
- Experimental details
For devices and machines used in research, indicate in parentheses (manufacturer, city, country). Are all formulas original? If not needed appropriate citations. Give the parameters of the workpiece material.
Comment 3:
- Measurement method of actual amplitude
Add grinding modes to the captions. For example, in figures 7.
Comment 4:
- Results and discussion
Add explanations to the captions: all symbols. For Figures 10, 11, 13, 14. This will facilitate understanding for readers without having to refer to the text of the article.
Comment 5:
It will be useful to add a section of Nomenclature in which to sign all the physical quantities and abbreviations encountered in the article. There are many physical quantities in the text and such a section will help to find the description of the necessary element.
For example,
ap : Depth of cut (µm)
LTUAG : Longitudinal-torsional coupled ultrasonic vibration assisted grinding
etc.
The article is interesting and written at a good scientific level. Authors should carefully study the comments and make improvements to the article step by step. After major changes can an article be considered for publication in the "Materials".
Round 2
Reviewer 1 Report
The required modifications are performed, except the one regarding the conventional process. However, the present form of the manuscript provides a suitable insight into torsional vibration effects, is of interest, and can therefore be published.
Reviewer 2 Report
The authors took into account all the comments. The article can now be published.